# Cannabidiol Inhibits Tumorigenesis in Cisplatin-Resistant Non-Small Cell Lung Cancer via TRPV2

**DOI:** 10.3390/cancers14051181

**Published:** 2022-02-24

**Authors:** Swati Misri, Kirti Kaul, Sanjay Mishra, Manish Charan, Ajeet Kumar Verma, Martin P. Barr, Dinesh K. Ahirwar, Ramesh K. Ganju

**Affiliations:** 1Department of Pathology, The Ohio State University, Columbus, OH 43210, USA; misri.2@osu.edu (S.M.); kirti.kaul@osumc.edu (K.K.); sanjay.mishra@osumc.edu (S.M.); manish.charan@osumc.edu (M.C.); ajeet.verma@osumc.edu (A.K.V.); 2Thoracic Oncology Research Group, Trinity St. James’s Cancer Institute, St. James’s Hospital, D08 W9RT Dublin, Ireland; mbarr@stjames.ie; 3School of Medicine, Trinity Translational Medicine Institute, Trinity College Dublin, D08 W9RT Dublin, Ireland; 4Comprehensive Cancer Center, The Ohio State University, Columbus, OH 43210, USA

**Keywords:** cisplatin resistance, TRPV2, cannabidiol, apoptosis, non-small cell lung cancer

## Abstract

**Simple Summary:**

Drug resistance is the key factor contributing to the therapeutic failure of lung cancer and the deaths related to lung cancer. Our study demonstrated that small molecular weight non-psychotropic phytochemical, cannabidiol (CBD), inhibits growth and metastasis of drug-resistant non-small cell lung cancer cells (NSCLC) cells in-vitro and in-vivo. We further discovered that CBD mediates its anti-cancer effects in part via an ion channel receptor, TRPV2, present on lung adenocarcinoma. Moreover, we showed that CBD induces apoptosis of cisplatin-resistant cells by modulating oxidative stress pathways. Overall, these studies indicate that CBD could be used as a promising therapeutic strategy in TRPV2 expressing cisplatin-resistant NSCLC.

**Abstract:**

Chemotherapy forms the backbone of current treatments for many patients with advanced non-small-cell lung cancer (NSCLC). However, the survival rate is low in these patients due to the development of drug resistance, including cisplatin resistance. In this study, we developed a novel strategy to combat the growth of cisplatin-resistant (CR) NSCLC cells. We have shown that treatment with the plant-derived, non-psychotropic small molecular weight molecule, cannabidiol (CBD), significantly induced apoptosis of CR NSCLC cells. In addition, CBD treatment significantly reduced tumor progression and metastasis in a mouse xenograft model and suppressed cancer stem cell properties. Further mechanistic studies demonstrated the ability of CBD to inhibit the growth of CR cell lines by reducing NRF-2 and enhancing the generation of reactive oxygen species (ROS). Moreover, we show that CBD acts through Transient Receptor Potential Vanilloid-2 (TRPV2) to induce apoptosis, where TRPV2 is expressed on human lung adenocarcinoma tumors. High expression of TRPV2 correlates with better overall survival of lung cancer patients. Our findings identify CBD as a novel therapeutic agent targeting TRPV2 to inhibit the growth and metastasis of this aggressive cisplatin-resistant phenotype in NSCLC.

## 1. Introduction

Lung cancer is the leading cause of cancer related deaths in the USA. In 2021, there were an estimated 131,880 deaths from this disease [1]. Non-small cell lung cancer (NSCLC) accounts for the highest percentage of lung cancers diagnosed. Access to the tumor and therefore surgical resection poses a great challenge due to structural and location complexity. As a result, cisplatin chemotherapy has remained a first line therapy for the majority of these patients. While most NSCLC patients initially demonstrate a clinical benefit from cisplatin-based chemotherapy, tumor recurrence eventually ensues due to the development of drug resistance, which has become a significant clinical challenge [2]. Drug resistance in cancer patients is often attributed to the emergence and expansion of cancer stem cells (CSCs), which have self-renewing properties and the ability to asymmetrically divide [3,4]. CSCs escape the cytotoxic effects of chemotherapy due to a number of mechanisms, including the dysregulation of drug transporters resulting in drug efflux and subsequent tumor relapse [5]. It is now well documented that such mechanisms also involve the activation of pro-survival and anti-apoptotic pathways [6]. Therefore, inducing apoptosis in cisplatin-resistant (CR) cells may be a promising strategy to target drug-resistant lung tumors.

Cannabidiol (CBD), a non-psychoactive phytochemical derived from the Cannabis plant, has been used as a form of pain management in cancer patients [7]. CBD has also been reported to inhibit growth, proliferation, angiogenesis, and metastasis of breast cancer and glioblastoma [8]. Recent studies have shown that intake of CBD oil increased the regression of tumors in lung adenocarcinoma patients in the absence of other interventions such as chemotherapy or diet [9]. However, CBD’s effect on drug-resistant cancer cells, in particular CR NSCLC, has not yet been reported, and its molecular mechanisms of action have not been fully characterized.

Transient receptor potential vanilloid-2 (TRPV2) is a member receptor belonging to the TRPV family that responds to osmolarity changes and noxious heat [10,11] It is a voltage gated ion channel and has been shown to regulate Ca^2+^ homeostasis during apoptosis [12,13]. TRPV2 has been shown to be highly expressed on triple-negative breast cancer (TNBC) [14]. TRPV2 activation by CBD significantly upregulates chemotherapy uptake and induces apoptosis in TNBC cells [14]. However, the role of TRPV2 in CR NSCLC is not known.

In the present study, we report the inhibitory properties of CBD on the growth and metastasis of CR NSCLC in vitro and in vivo compared to cisplatin treatment. Our data show that CBD induces apoptotic signals, together with increased reactive oxygen species in CR cells in a TRPV2-dependent manner. High expression of TRPV2 is found in NSCLC cell lines and in lung adenocarcinoma patients, where it is associated with poor overall survival. This study provides novel insights into the anti-tumor effects mediated by CBD in CR NSCLC.

## 2. Materials and Methods

### 2.1. Cell Lines

H460 (large cell carcinoma) and A549 (adenocarcinoma) cells were purchased from the ATCC (Manassas, VA, USA) from which cisplatin-resistant (CR) sublines were generated [15]. H460-parental (PT) and H460-CR cells were cultured in RPMI-1640 supplemented with 1% penicillin-streptomycin (sigma Aldrich, p4333, St. Louis, MO, USA) and 10% fetal bovine serum (FBS) (Sigma Aldrich, PMS-013-B). CR cells were maintained in 6.6 µM cisplatin (Sigma Aldrich). A549 PT and A549-CR were cultured in DMEM (Corning, Corning, NY, USA) supplemented with 1% penicillin-streptomycin (Sigma Aldrich, p4333) and 10% fetal bovine serum (FBS) (sigma Aldrich, PMS-013-B), while A549-CR cells were cultured in 11.6 µM cisplatin (Sigma Aldrich). All cell lines were maintained as monolayer cultures at 37 °C in a humidified incubator at 5% CO_2_.

### 2.2. Cell Viability

Cell viability assays was performed using Presto Blue dye™ (ThermoFisher, Carlsbad, CA, USA) as per manufacturer’s instructions. Briefly, 2 × 10^3^ cells were seeded per well in 90 μL of medium in 96-well plates and serum starved for 4 h before treatment. Cells were treated with increasing concentrations of cisplatin and CBD for 48 h. At the end of this period, Presto Blue™ solution was added to each well and incubated for 30 min. Fluorescence was measured at 570 nm and 600 nm using the BioRad (Hercules, CA, USA) microplate reader. Dose–response graphs were constructed, from which IC_50_ concentrations were determined for each cell line.

### 2.3. Colony Formation

Cells (1 × 10^3^) were seeded in each well of a 6-well plate suspended in complete growth media for approximately 16 h, after which time serum free media was used for 6 days for treatment with vehicle control (PBS), CBD, or cisplatin. Colonies were then stained with Giemsa stain containing alcohol for fixation and counted manually by using Leica brightfield microscope (Wetzlar, Germany).

### 2.4. Apoptosis

Percent apoptosis was determined using APC-labeled Annexin-V and 7-AAD (BD Biosciences, Franklin Lakes, NJ, USA). Following treatment with vehicle control, cisplatin, or CBD for 48 h, drug-resistant H460-CR and A549-CR NSCLC cells were stained with Annexin-V-APC and 7-AAD according to the manufacturer’s instructions. 1 × 10^4^ cell events were recorded to determine the percentage of live, apoptotic, and necrotic cells using flow cytometry (BD Fortessa, Columbus, OH, US) and analyzed using FlowJo software.

### 2.5. Western Blot

Cell lysates were prepared using RIPA lysis buffer and resolved on NuPAGE 4–12% gradient precast gels (Invitrogen, Waltham, MA, USA) and transferred onto 0.45 μm nitrocellulose membranes (BioRad, Hercules, CA, USA). Membranes were blocked for 1 h in 5% blocking-grade milk and probed with specific primary antibodies for cleaved-caspase-3, 9, snail, nanog, vimentin (CST, Danvers, MA, USA), and NRF-2 (Santa Cruz biotechnology, Dallas, TX, USA) using a 1:1000 dilution. HRP conjugated secondary goat anti-mouse IgG or anti-rabbit IgG (ThermoFisher, Carlsbad, CA, USA) (1:5000) was used and membranes were developed using a chemiluminescent substrate (Sigma Aldrich, St. Louis, MO, USA). β-actin (Santa Cruz biotechnology, Dallas, TX, USA) was used as a loading control.

### 2.6. ROS Generation

Cellular ROS Assay kit (Red) ab186027 (Abcam, Cambridge, UK) uses a Reactive Oxygen Species (ROS) sensor to quantify ROS in live cells. The red dye used in the ROS assay protocol is cell-permeable and generates red fluorescence when it reacts with ROS. Cells were treated with either vehicle control, cisplatin, or CBD after adding the red dye and ROS generation was recorded on the SpectraMax ID5 instrument (San Jose, CA, USA) at different time points and 1 h time point is used in the figures.

### 2.7. Sphere Formation

H460-CR and A549-CR cells (1 × 10^3^) were cultured in Mammocult media (100 µL) and suspended in ultralow attachment 96 well plates for 7 days with vehicle control, CBD, or cisplatin, during which time cancer stem cell spheres were formed. The spheres were counted manually. In addition, H460-CR and A549-CR cells (1 × 10^3^) were cultured in Mammocult media using ultralow attachment 6mm plates and grown for 10 days to allow for sphere formation, after which time spheres were trypsinized to make a single cell suspension. These were stained for cell surface markers CD133 and CD44 (Biolegend, San Diego, CA, USA) and 1 × 10^4^ cell events were recorded to determine the percent population of double-positive cells using flow cytometry (BD Fortessa, Columbus, OH, USA). Analysis was carried out using FlowJo software.

### 2.8. Immunofluorescence

1 × 10^6^ cells were seeded in the chambered slides (ThermoFisher, Carlsbad, CA, USA) and left for overnight attachment, followed by the treatment of CBD or Cisplatin or Vehicle control for 6h. Then slides were fixed with 4% formaldehyde and the blocked with 5% normal serum 0.3% tritonx100 for 1 h. After blocking slides were incubated with primary Cyt-c (1:100) (CST, Danvers, MA, USA) primary antibody overnight at 4 degrees. After 3 gentle washes with PBS slides were incubated with HRP conjugated secondary antibody (1:500) for 1h in dark followed by 3 washes with PBS and were mounted with antifade mounting media (Vectashield, Burlingame, CA, USA).

### 2.9. Animal Studies

Six-week-old female NSG mice were obtained from the OSU core facility. Mice (*n* = 5/group) were subcutaneously injected with H460-CR cells (1 × 10^6^) in the right flank. Once tumors became palpable, mice were treated with vehicle control (PBS), CBD (10 mg/kg), or cisplatin (5 mg/kg) once a week for 4 weeks via intraperitoneal (i.p) route. Mice were observed weekly, and tumor volumes were measured using vernier caliper until tumors reached 2000 (mm^3^) or body score was less than 3 in accordance with IACUC protocol. Tumors and lungs were harvested from all mice. All experiments were approved by the Institutional Animal Care and Use Committee (IACUC) of the Ohio State University and animals were housed as per University Laboratory Animal Resources (ULAR) guidelines.

### 2.10. Hematoxylin and Eosin Staining

Hematoxylin and Eosin staining was performed by the core facility at the Department of Pathology, OSU. Briefly, tissue sections were deparaffinized, then rehydrated with decreasing concentrations of alcohol. After rehydration, sections were stained with hematoxylin nuclear stain and eosin as a counterstain [16].

### 2.11. Free Intracellular Calcium

Cells (3 × 10^6^) were seeded in 96-well clear bottom plates in complete media. Calcium indicator, Calcium Green-1 AM, was added to the media at a final concentration of 5 µM. After adding the calcium indicator, cells were treated with vehicle control, CBD, a TRPV2 inhibitor- Tranilast (TLS) (10 µM) (Sigma Aldrich, St. Louis, Miss, USA), or TLS + CBD. To evaluate fluorescent intensities, fluorescence readings were recorded on a microplate reader (Biotek Synergy H1, Winooski, VT, USA) for 5 min at 1 min intervals. Fluorescent intensities were also recorded following the addition of ionomycin (5 µM) as the positive control.

### 2.12. Statistics

Statistical analyses were performed using Prism software 9.2 (Graph Pad Software Inc., San Diego, CA, USA). Unpaired student’s *t*-test was used for comparing two groups and ANOVA was used for comparing between more than two groups. Statistical significance was noted in the figures * *p* < 0.01, ** *p* < 0.001, *** *p* < 0.001, **** *p* < 0.0001, and ns was non-significant. All figures are represented as Mean ± S.D for three independent experiments.

## 3. Results

### 3.1. CBD Reduces Viability of CR NSCLC Cells

We first evaluated the effect of increasing concentrations of cisplatin and CBD on the cell viability of NSCLC parental (H460-PT and A549-PT) and corresponding cisplatin-resistant cell lines (H460-CR and A549-CR). Cisplatin treatment reduced the cell viability of H460 PT (IC_50_ 27.8 µM) and A549 PT (IC_50_ 40.2 µM) cells at much lower dose compared to cisplatin-resistant H460-CR (IC_50_ 73.2 µM) and A549-CR (IC_50_ > 150 µM) cells (Figure 1A,B). However, CBD treatment had drastic inhibitory effect on the viability of both PT cells (H460- IC_50_ 15.8 µM and A549- IC_50_ 16 µM) and cisplatin-resistant cells (H460-CR- IC_50_ 14.2 µM and A549-CR- IC_50_ 15.9 µM) (Figure 1C,D). We have analyzed the effect of CBD in combination with cisplatin, however we did not find any synergistic effects on cell viability. We have observed significant drastic effects with CBD treatment alone.

CR NSCLC cells were evaluated for their survival ability in-vitro in response to cisplatin or CBD using the colony formation assay. Cells were diluted to form single cell suspension with as low as 2 × 10^3^ cells per well and were grown for 6 days in serum starved conditions with CBD or cisplatin treatment. Analysis of the number of colonies showed that CBD treatment significantly reduced the surviving fraction of colonies in H460-CR and A549-CR cell lines compared to vehicle control. However, while cisplatin treatment did not inhibit colony formation of these cells. (Figure 1E,F), Of interest, cisplatin significantly induced the formation of colonies relative to cells treated with vehicle control (Figure 1E,F).

### 3.2. CBD Induces Apoptosis in CR NSCLC

We next evaluated if reduced viability in CBD-treated CR NSCLC cells was linked to increased apoptosis. Indeed, CBD treated CR NSCLC cells show increased number of apoptotic cells compared to vehicle control or cisplatin (Figure 2A,B). We further validated the effect of CBD in inducing apoptosis in CR NSCLC cells by analyzing the expression of various apoptosis markers. Western blot analysis showed that CBD-treated CR NSCLC cells had significantly higher expression of the apoptosis markers, cleaved caspases 3 and 9 compared to VC or cisplatin treatments (Figure 2C–F and Appendix A). Release of cytochrome c is known to activate apoptosis by enhancing caspases 3 and 9. We therefore evaluated cytochrome-c levels in these cells and observed that CBD-treated H460 CR and A549-CR cells had higher levels of cytochrome c release compared to VC or cisplatin treatment (Figure 2G,H). These studies suggest that CBD induces apoptosis of CR NSCLC cells.

### 3.3. CBD Activates Reactive Oxygen Species (ROS) and Suppresses NRF-2

Drug-resistant cells have been shown to have high levels of antioxidant enzymes and other factors responsible for the production of antioxidants [17,18]. Emerging evidence suggests that cisplatin-resistant cells have low levels of intracellular ROS [19]. In addressing whether ROS levels are different in CBD, VC, or cisplatin-treated CR NSCLC cells, ROS levels were found to be significantly higher in CBD-treated H460-CR and A549-CR cells compared to VC or cisplatin-treated cells (Figure 3A,B and Appendix A). The NRF-2 transcription factor is a master regulator of ROS levels [20,21]. Endogenous NRF-2 expression was higher in H460-CR and A549-CR cells relative to their PT counterparts (Figure 3C,F and Appendix A). Further analysis demonstrated that CBD treatment decreased NRF-2 expression in CR NSCLC cells compared to VC or cisplatin treatments (Figure 3G–J and Appendix A). These data suggest that CBD-mediated apoptosis of CR NSCLC cells is regulated via a ROS/NRF-2 pathway.

### 3.4. CBD Inhibits Sphere Formation and Stemness Gene Expression

Cancer stem cells (CSCs) play a vital role in the development of drug resistance. Spontaneous mutations and prolonged exposure to cisplatin enables CSCs to surpass the cytotoxicity of cytotoxic agents and develop anchorage independent growth that leads to sphere formation. Therefore, we evaluated sphere formation ability of CBD-treated CR NSCLC cells and observed that CBD treatment results in significantly fewer spheres in H460-CR and A540-CR cells compared to VC or cisplatin-treated cells (Figure 4A,B). To further evaluate the effect of CBD treatment on the cancer stem, cell phenotypic cells were assessed for CD44^+^CD133^+^ CSC markers. CBD-treated CR NSCLC cells had significantly lower numbers of CD44^+^ and CD133^+^ cells (H460-CR; 41.73% vs. A549-CR; 81.70%) in comparison to VC controls (H460; 91% vs. A549; 95.877%) (Figure 4C,D). We further evaluated the effects of CBD treatment on a panel of CSC markers and found that CBD downregulated protein expression of Snail, Nanog, and Vimentin were significantly in both H460 (Figure 4E,F and Appendix A) and A549 (Figure 4G,H and Appendix A) NSCLC cells, highlighting a role for CBD in downregulating stem cell properties in CR cells.

### 3.5. CBD Inhibits Tumor Growth and Metastasis of Cisplatin-Resistant Cells In Vivo

To confirm these in vitro data, we evaluated the ability of CBD to inhibit the growth and metastasis of CR NSCLC cells in vivo. CBD treatment significantly inhibited tumor growth of H460-CR xenografts in NSG mice (Figure 5A–C) compared to VC-treated tumors. Interestingly, there was no significant reduction in tumor volume or tumor weight in response to treatment with cisplatin (Figure 5A–C). IHC staining showed an increase in cleaved caspase-3 in CBD-treated tumors, with no observed changes in response to cisplatin (Figure 5D,E). The effect of CBD treatment on tumor metastases was also examined. Lungs were harvested from control (VC), cisplatin or CBD-treated mice and analyzed for micro-metastases by H&E staining. As shown in Figure 5F–H, CBD treatment significantly reduced micro-metastatic nodules compared to VC. However, cisplatin treatment showed a significant increase in macro-metastatic nodules compared to control (VC) mice (Figure 5F–H). These data suggest an anti-tumor growth effect of CBD in H460-CR cells.

### 3.6. CBD Interacts with TRPV2 to Induce Apoptosis

CBD has previously been shown to act through TRPV2 in breast cancer. To evaluate the clinical relevance of TRPV2 expression in lung cancer, gene expression analysis was performed using The Cancer Genome Atlas (TCGA) dataset, where it was found that patients with lung adenocarcinoma had higher expression of TRPV2 compared to patients with squamous cell carcinoma (Figure 6A). Further, we used a KM plotter to analyze overall survival of patients based on TRPV2 expression and observed that lung cancer patients with higher expression of TRPV2 has a significantly better overall survival (*p* = 4.3 × 10^−6^) compared to patients with low TRPV2 expression (Figure 6B) [22]. We further determined if CBD mediates its effects through TRPV2 by analyzing the effects of CBD on apoptosis of cisplatin resistant H460 and A549 NSCLC cells in the presence or absence of the TRPV2 inhibitor. Tranilast (TLS). To determine the dose of TLS, which reverses the effect of CBD without initiating cell death, we subjected cells to pre-treatment with different doses of TLS (TRPV2 inhibitor) followed by CBD treatment and observed that the best dose was 10 uM of TLS for reversing the effect of CBD (data not shown). TLS treatment significantly abrogated CBD-induced apoptosis in CR NSCLC cells (Figure 6C,D) in addition to CBD-induced activation of the apoptosis markers, caspase 3 and 9 (Figure 6E–H and Appendix A). Homeostasis of intracellular Ca^2+^ is essential for cisplatin-induced apoptosis in ovarian cancer [23], and it is suggested in some studies that cisplatin resistance is supported by Ca^2+^ efflux through TRPV2. In cisplatin-resistant NSCLC cells, CBD treatment increased intracellular Ca^2+^ levels that were abrogated in the presence of TLS (Figure 6I,J). However, TLS alone had no significant effect on apoptosis or intracellular Ca^2+^ levels. These findings suggest that CBD may mediate its anti-tumor effects through TRPV2 [14].

## 4. Discussion

Among the different types of lung cancers, NSCLC is the most common and comprises of approximately 80% of all lung cancers. Until more recently, since the advent of immune therapies, cisplatin-doublet chemotherapy remained a standard treatment of care for NSCLC patients with advanced disease. Unfortunately, relapse occurs in many patients due to the development of acquired drug resistance during treatment, which is a significant challenge for oncologists treating these patients. Thus, there is an urgent need for identifying novel targets and developing new therapeutic strategies to overcome this resistance phenotype in NSCLC. Here, we report a novel use for CBD in inhibiting the tumorigenic properties of cisplatin-resistant (CR) NSCLC cells in vitro. Furthermore, our in vivo studies show that treatment with CBD significantly inhibits the growth and metastasis of human CR NSCLC xenografts. CBD is derived from cannabis extracts and does not show any psychotropic activity or other adverse effects [24,25,26]. It has been approved by the FDA and several clinical trials using CBD as a single agent in solid tumors have been documented in cancer [27,28] and pediatric epilepsy [29,30]. Notably, oral administration of CBD (600–800 mg per day) has been shown to be safe and well-tolerated in clinical trials conducted in healthy subjects [31,32,33].

CSCs has been shown to play an important role in cancer cell survival, growth, and metastasis [34]. It is now well documented in vitro and in vivo that CSCs play a key role in the development of drug resistance in different cancer types [35,36,37]. Therefore, it is critical to identify novel drugs that can target CSCs within the tumor cell population. In the present study, our data show that CBD reduces CSC populations in CR cell lines, thereby adding to its overall tumorigenic effects in NSCLC.

CBD has been shown to induce apoptosis in different tumors. However, it remains unclear as to the exact mechanism(s) by which CBD induces apoptosis in drug-resistant cells, in particular cisplatin-resistant lung cancer cells. We have shown that CBD inhibits the growth of CR NSCLC cells via the induction of apoptosis and a parallel reduction in NRF-2 expression and increase in ROS generation in these drug-resistant cells. Previous studies have shown that the induction of cellular ROS is an important mechanism of chemotherapy -induced cell death. However, chemotherapy-resistant cancer cells enhance NRF-2-mediated molecular signaling to counteract ROS and escape apoptosis [38,39,40]. Our findings highlight a critical role for CBD in reducing NRF-2 expression, which in turn enhances the production of cellular ROS, providing a novel insight to the role of CBD in overcoming this pro-apoptotic pathway in CR NSCLC.

CBD has been shown to mediate its effects through the TRPV2 ion channel. Recent reports have demonstrated that CBD can directly interact with TRVP2 through a hydrophobic pocket located between S5 and S6 helices of adjacent subunits [41]. Data from cryo-electron microscopy of apo and CBD-bound states of the TRPV2 channel show that the S4-S5 linker is critical for channel gating upon CBD binding [41]. We have found that TRPV2 is more highly expressed in lung adenocarcinomas compared to squamous cell carcinomas. Furthermore, our studies indicate that higher expression of TRPV2 correlates with improved overall survival [42], while CBD induces apoptosis in CR NSCLC cells in vitro through TRPV2, where inhibition of TRPV2 using Tranilast abrogates CBD-mediated intracellular Ca^2+^ levels and cell apoptosis. Recent studies have shown that TRPV2 promotes H_2_O_2_-induced oxidative stress and cytotoxicity in human hepatoma cells [43]. The activation of TRPV2 through CBD has also been shown to induce apoptosis in glioblastoma cells without causing any considerable cytotoxicity on normal astrocytes [43]. These studies suggest that TRPV2 could be used as a prognostic marker for NSCLC patients in response to treatment, with CBD as a novel therapeutic agent in the context of cisplatin NSCLC.

Taken together, these findings demonstrate for the first time, to our knowledge, that CBD treatment of cisplatin-resistant NSCLC cells inhibits tumor growth and metastasis. Mechanistic studies in vitro indicate that CBD induces apoptosis in these drug-resistant lung cancer cells via a novel mechanism involving NRF2 and ROS-mediated signaling pathways and cancer stemness. Importantly, our data show that CBD mediates these effects through TRPV2, which is highly expressed in lung adenocarcinomas. While further exploratory studies are warranted, these findings indicate the potential use of CBD as an innovative and novel therapeutic strategy in cisplatin-resistant NSCLC patients.

## 5. Conclusions

In this study, we show that CBD is a potent small molecular weight FDA-approved drug with anti-tumor effects in drug-resistant NSCLC. These effects are mediated, at least in part, via TRPV2, and enhance the inductions of tumor cell apoptosis by modulating oxidative stress pathways and stem cell properties in cisplatin-resistant lung cancer cells. These data support pre-clinical studies to further investigate the potential use of CBD cisplatin-resistant NSCLC.

## Figures and Tables

**Figure 1 cancers-14-01181-f001:**
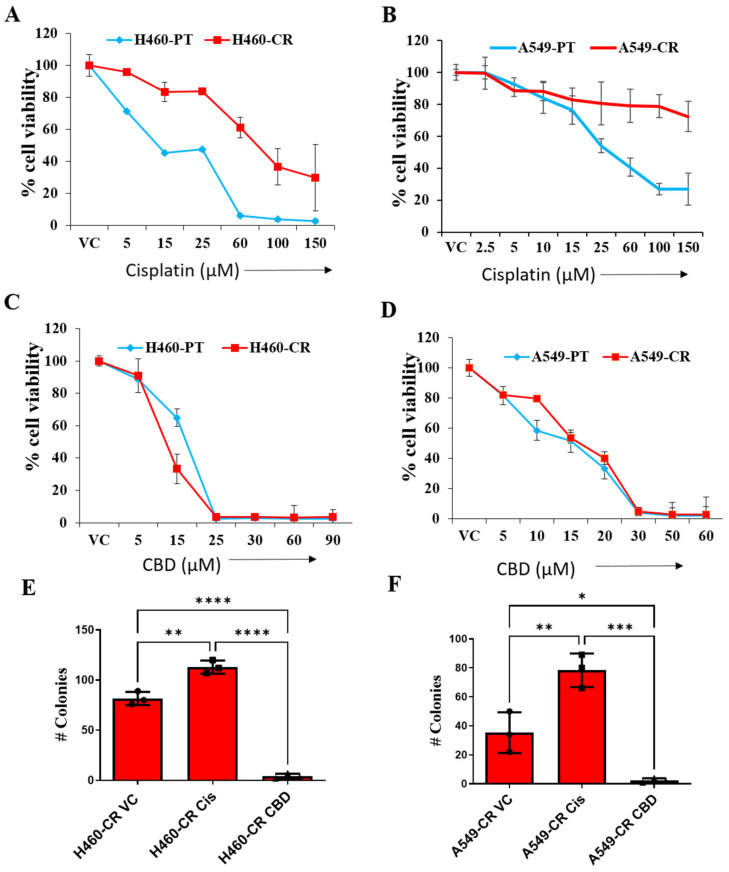
CBD decreases viability of CR NSCLC cells. NSCLC parental controls (H460-PT, A549-PT) or drug-resistant (H460-CR, A549-CR) cells were treated with increasing concentrations of cisplatin (**A**,**B**) or CBD (**C**,**D**) for 48 h. PBS was used as vehicle control (VC). Graphs represent precent cell viability analyzed using presto blue dye. (**E**,**F**) CR NSCLC cells were treated with PBS vehicle control (VC) or CBD or cisplatin and the cells were analyzed for the number of colonies. Data are represented as Mean ± S.D (* *p* < 0.01, ** *p* < 0.001, *** *p* < 0.001, **** *p* < 0.0001 using one-way ANOVA ns, non-significant).

**Figure 2 cancers-14-01181-f002:**
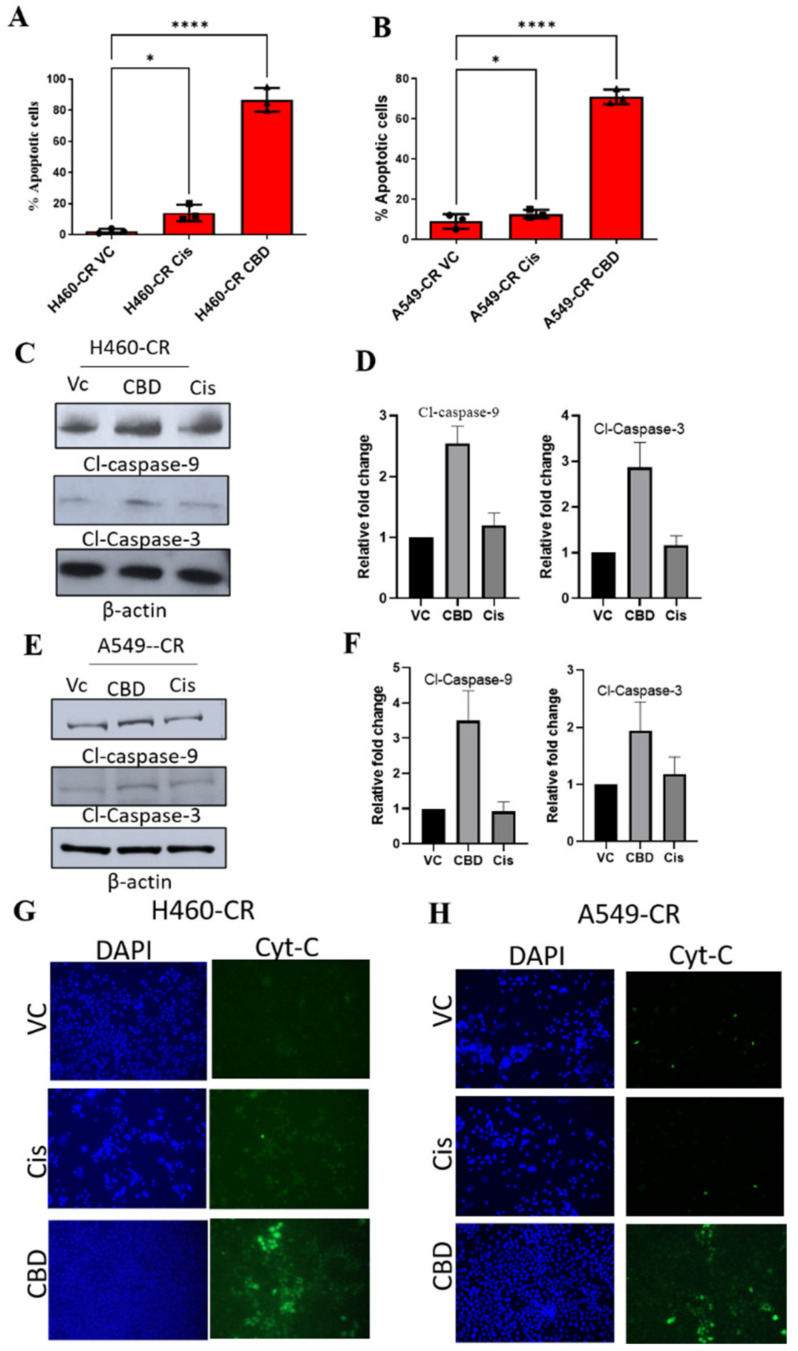
CBD-mediated induction of apoptosis in CR NSCLC cells. (**A**) H460-CR or (**B**) A549-CR cells were treated with VC, cisplatin, or CBD. Percentage apoptotic cells were analyzed using Annexin-V and 7-AAD staining and analyzed by flow cytometry analysis. (**C**,**E**) H460-CR or A549-CR cells treated with VC or CBD or Cisplatin and cell lysates were evaluated for caspase 3 and caspase 9 apoptotic markers by Western blot analysis. β-actin was used as loading control. (**D**,**F**) Graphs represents densitometric ratio in fold change for H460-CR or A549-CR respectively. (**G**), H460-CR or (**H**) A549-CR cells were treated with VC, cisplatin, or CBD, and cytochrome-c release was measured by immunofluorescence, all the images were taken at 10X magnification. Data are represented as Mean S.D (* *p* < 0.01, **** *p* < 0.0001 using one-way ANOVA).

**Figure 3 cancers-14-01181-f003:**
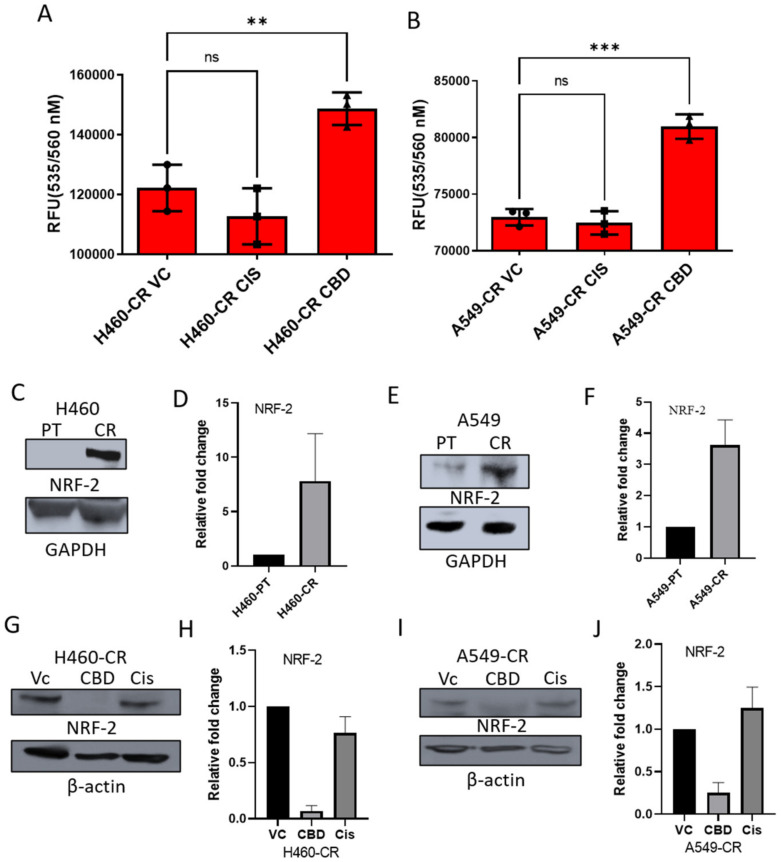
CBD induces Reactive Oxygen Species (ROS) and suppresses NRF-2 in CR NSCLC (**A**) CR-H460 or (**B**) CR-A549 cells were treated with VC, cisplatin, or CBD and analyzed for ROS production after 1 h. (**C**) Cell lysates were collected from H460-PT/CR and (**E**) A549-PT/CR cells and NRF-2 protein examined at baseline by Western blot analysis. (**D**,**F**) densitometric ratio represented in fold change, respectively. Similarly, NRF-2 expression was assessed in response to treatment with VC, CBD, or cisplatin in (**G**) H460-CR and (**I**) A549-CR cells. GAPDH was used as a loading control. (**H**,**J**) represents densitometric ratio in fold change, respectively. Data are represented as Mean ± S.D (** *p* < 0.001, *** *p* < 0.001, ns as non-significant using one-way ANOVA).

**Figure 4 cancers-14-01181-f004:**
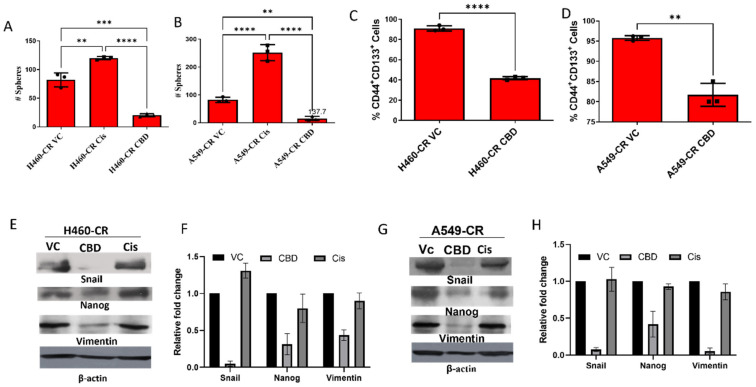
CBD inhibits sphere formation and stem-like properties in cisplatin-resistant NSCLC cells. (**A**) H460-CR and (**B**)A549-CR cell lines were treated with vehicle control (VC), cisplatin (27 μM), or CBD (15.8 µM) for 7 days and the number of spheres quantified. (**C**) H460-CR and (**D**) A549-CR cell lines were cultured and treated with CBD to generate spheres, after which time cell surface marker expression of CD44^+^CD133^+^ was examined by flow cytometry. (**E**) H460-CR and (**G**) A549-CR cell lysates were collected from CBD and cisplatin-treated cells and protein expression of Snail, Nanog, and Vimentin was examined by Western blotting. β-actin was used as a loading control (**F**,**H**, respectively) represents densitometric ratio in fold change. Data are represented as Mean ± S.D (** *p* < 0.001, *** *p* < 0.001, **** *p* < 0.0001, ns; non-significant). Statistical comparisons between two groups were carried out using Students *t*-test, while a one-way ANOVA was used for more than two groups.

**Figure 5 cancers-14-01181-f005:**
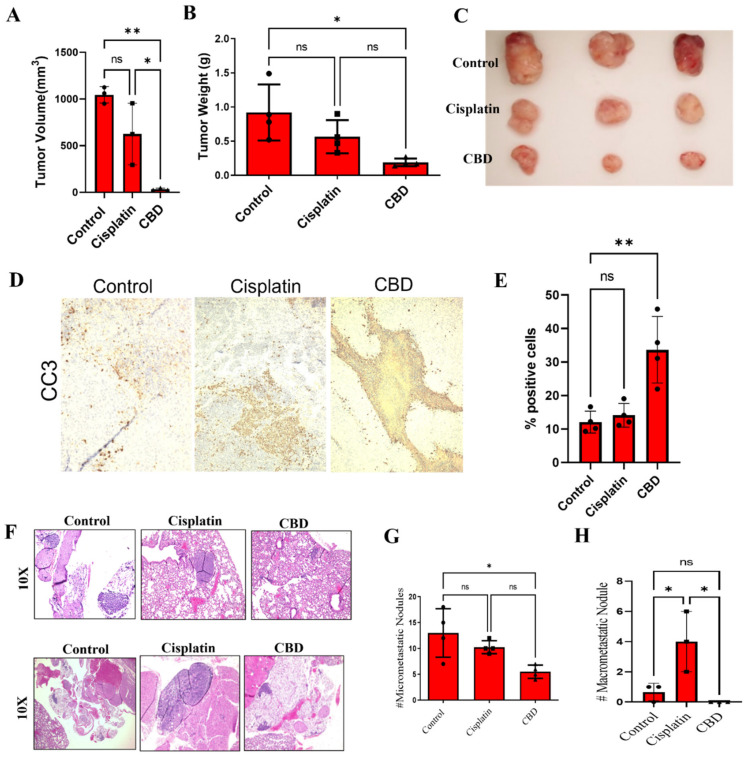
CBD inhibits CR NSCLC growth and metastasis in vivo. (**A**–**C**) H460-CR cells were injected subcutaneously into the right flank of nude mice (*n* = 5). Palpable tumors were treated with control (PBS) or CBD (10mg/kg body weight) or cisplatin (5 mg/kg body weight) for 4 weeks. (**A**) The tumor volume was measured externally at the end. (**B**) Weight of harvested tumors, and (**C**) the tumors were harvested, and representative images are presented. (**D**) The tumor sections were processed and immunostained for cleaved-caspase-3 (CC3) and images were taken at 10X magnification. (**E**) The CC3 staining images from (**D**) were quantified for CC3 % positive cells. (**F**–**H**) The lungs harvested from (**A**) were processed and H&E stained. (**F**) Representative H&E images of the lungs were harvested at 10x magnification. Upper and lower panel represents micro- and macro-metastasis, respectively. (**G**) Quantification of micro- and (**H**) macro-metastasis using ImageJ analysis. * is *p*-value < 0.01, ** *p* < 0.001, and ns is non-significant using one-way Anova.

**Figure 6 cancers-14-01181-f006:**
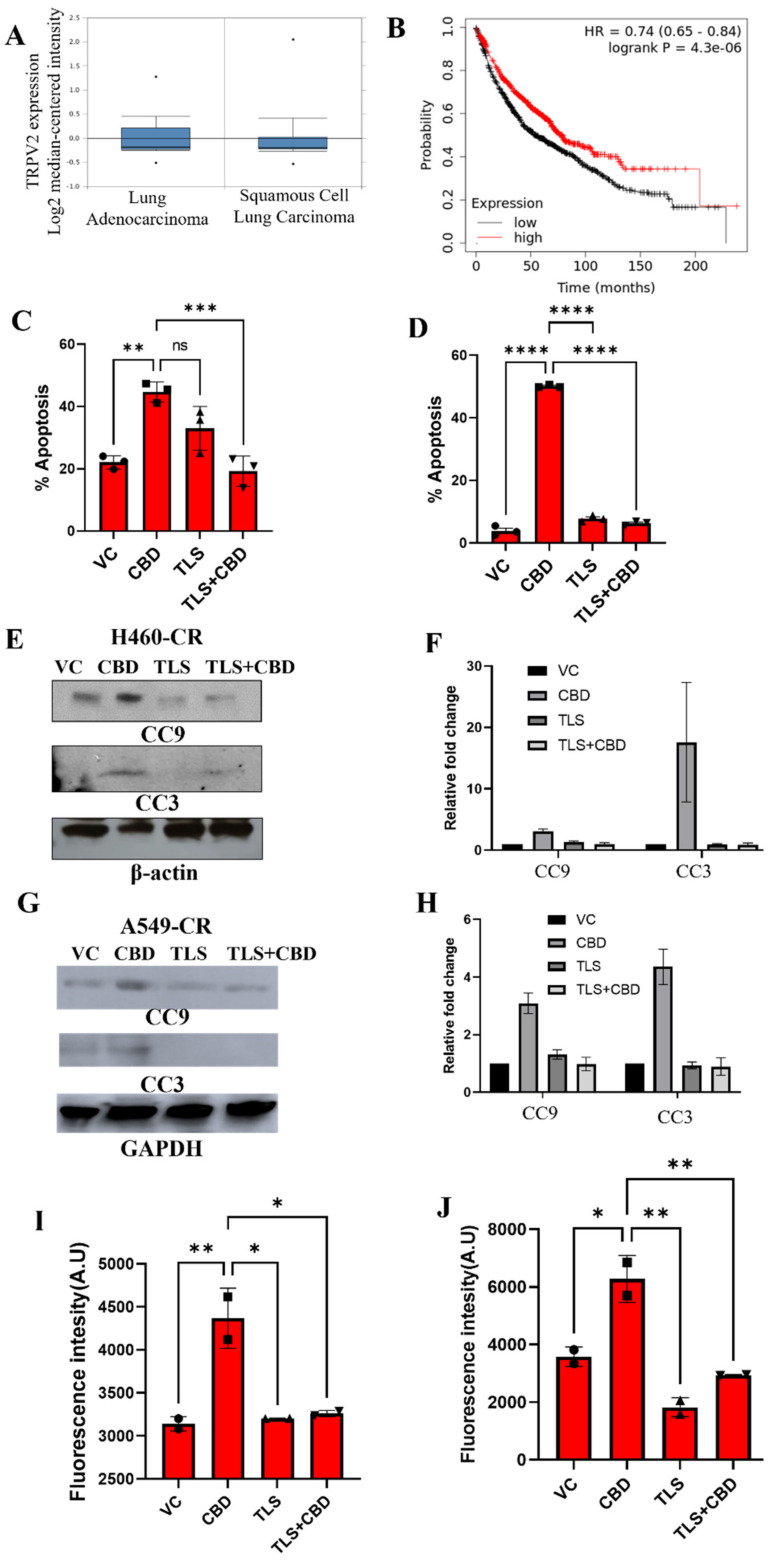
CBD interacts with TRPV2 channel to induce apoptosis. (**A**) Analysis of TRPV2 expression in lung adenocarcinoma (*n* = 63) and squamous cell carcinoma (*n* = 75) patient tumors using TCGA dataset. (**B**) The association between TRPV2 expression (low vs high) and overall survival of lung cancer patients was analyzed using KM plotter. (**C**) H460-CR and (**D**) A549-CR cell lines were treated with VC or CBD in the presence or absence of TLS. The percentage apoptosis was evaluated using annexin/7-AAD staining and flow cytometry. Cell lysates from (**E**) H460-CR and (**G**) A549-CR cell lines treated with VC or CBD in the presence or absence of TLS were analyzed for the expression of apoptosis markers, cleaved caspase 9 (CC9), and cleaved caspase 3 (CC3) by Western blot analysis using beta-actin and GAPDH as loading control. (**F**,**H**) Graphs represents densitometric ratio in fold change for (**E**,**G**), respectively. (**I**) H460-CR and (**J**) A549-CR cell lines were treated with VC or CBD, the presence or absence of TLS, and the free intracellular calcium levels were analyzed by estimating fluorescence intensity of calcium green-1-AM staining. Data are represented as Mean ± S.D (* *p* < 0.01, ** *p* < 0.001, *** *p* < 0.001, **** *p* < 0.0001, ns; non-significant). Statistical comparisons between two groups were carried out using Students *t*-test, while a one-way ANOVA was used for more than two groups.

## Data Availability

Data sharing not applicable. No new data were created or analyzed in this study.

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
