# Peer review of "Cannabidiol Inhibits Tumorigenesis in Cisplatin-Resistant Non-Small Cell Lung Cancer via TRPV2"

_cancers, 2022, doi:10.3390/cancers14051181_

Round 1

Reviewer 1 Report

The manuscript describe a sound and linear experiment to disentangle a potential mechanism for Platinum resistance in NSCLC.

Some minor issue regarding the English style and some spell. 

In the Introduction should be better referenced the concept that resistance could be mostly ascribed to CSC  and that in this population of cells drug transport is the main mechanism responsible of the phenotype. It would also benefit from hype reduction on this point. 

Author Response

As requested by the Editor and based on the reviewers’ comments below, we have amended our manuscript in accordance with these suggestions in this most updated revised version. We trust that these minor corrections are now adequate to warrant publication in the journal, Cancers.

Reviewer 3

Comments and Suggestions for Authors

The manuscript describes a sound and linear experiment to disentangle a potential mechanism for Platinum resistance in NSCLC.

Some minor issue regarding the English style and some spell.

In the Introduction should be better referenced the concept that resistance could be mostly ascribed to CSC and that in this population of cells drug transport is the main mechanism responsible of the phenotype. It would also benefit from hype reduction on this point.

Response: We are very grateful to Reviewer 3 for these kind comments and suggestions. The manuscript has been updated to ensure that any errors in spelling or grammar have been corrected. We have now added additional references supporting a role for CSCs in drug resistance.

Reviewer 2 Report

The author mentioned TRVP2 pore blocker - Tranilast,  had not shown major effect on intracellular calcium, and CBD itself may be involved in enhancing anti-cancer effects via TRVP2. 

Given this is an important part, readers might want any evidence with references/literature to substantiate this finding. 

Author Response

As requested by the Editor and based on the reviewers comments below, we have amended our manuscript in accordance with these suggestions in this most updated revised version. We trust that these minor corrections are now adequate to warrant publication in the journal, Cancers.

Reviewer 2

Comments and Suggestions for Authors

The author mentioned TRVP2 pore blocker - Tranilast, had not shown major effect on intracellular calcium, and CBD itself may be involved in enhancing anti-cancer effects via TRVP2.

Given this is an important part, readers might want any evidence with references/literature to substantiate this finding.

Response:  We thank Reviewer 2 for this very relevant comment. We used a lower dose of tranilast (10µM) which did not initiate any cell death signaling but however, reversed these CBD-mediated effects. We have included a statement in section 3.6 of the revised manuscript to highlight this.

Reviewer 3 Report

The paper under review is a translational investigation into the properties of a small molecule in overcoming resistance in cells exposed to cisplatin. Previous work has indicated that cancer stem cells play a significant role in the induced resistance of cisplatin and Cannabidiol has been reported as being able to inhibit cell proliferation in NSCLC. In total the script is well prepared with a logical flow and slightly above average English competence. The results demonstrate are unambiguous and clearly demonstrated in corresponding figures (except for figure 6 which is missing), which are off good quality. In all I found the paper to be easy to review and interesting to read.

Questions

  1. I miss the combination treatments; can you achieve resistant cell lines when combining both cisplatin and cannabidiol? You have presented the whole paper as a treatment for patients relapsing after cisplatin treatment via the stem cell resurgence. The results presented are clear in that cannabidiol definitely has future role in the treatment of NSCLC but you do not indicate whether this would be a preventative treatment (i.e. pre or combination treatment) or supplementary post cisplatin treatment. It would be interesting to at least comment on this.

Very minor irritations and “petty comments”

  1. Line 49 – my pdf from cancers produced a font change in the part of the text to large, bold.
  2. Reference 15 - I fail to see the connection of reference 15 to the H&E staining. Also subtitle of the section should not be an abbreviation, I know it is common when referring to such a gold standard technique but should be explained a little more fully in publications
  3. For your information the special character “µ” is freely available in most fonts using alt 230. It has been used several times before in the text so I see no reason for using “uM” as a substitute for µM.
  4. Line 155 – Tranilast – where did you get it and what concentration did you use, also the citations you use for TRPV2 in the introduction should at the very least include!

Perálvarez-Marín A, Doñate-Macian P, Gaudet R. What do we know about the transient receptor potential vanilloid 2 (TRPV2) ion channel?. FEBS J. 2013;280(21):5471-5487. doi:10.1111/febs.12302

  1. Line 226 – grammar mistake should “maybe” not “may by”
  2. Figure 6 is missing from my pdf printout, the caption is present though
  3. Line 385 Sentence doesn’t quite make sense, I think you left out an “in”

Author Response

As requested by the Editor and based on the reviewers comments below, we have amended our manuscript in accordance with these suggestions in this most updated revised version. We trust that these minor corrections are now adequate to warrant publication in the journal, Cancers.

Reviewer 1

The paper under review is a translational investigation into the properties of a small molecule in overcoming resistance in cells exposed to cisplatin. Previous work has indicated that cancer stem cells play a significant role in the induced resistance of cisplatin and Cannabidiol has been reported as being able to inhibit cell proliferation in NSCLC. In total the script is well prepared with a logical flow and slightly above average English competence. The results demonstrate are unambiguous and clearly demonstrated in corresponding figures (except for figure 6 which is missing), which are off good quality. In all I found the paper to be easy to review and interesting to read.

Questions

  1. I miss the combination treatments; can you achieve resistant cell lines when combining both cisplatin and cannabidiol? You have presented the whole paper as a treatment for patients relapsing after cisplatin treatment via the stem cell resurgence. The results presented are clear in that cannabidiol definitely has future role in the treatment of NSCLC, but you do not indicate whether this would be a preventative treatment (i.e. pre or combination treatment) or supplementary post cisplatin treatment. It would be interesting to at least comment on this.

Response: We thank Reviewer 1 for these constructive comments in relation to treatments with cisplatin and cannabidiol. We have previously examined the effects of CBD in combination with cisplatin on cell viability. However, we did not find any synergistic effects in response to this combination treatment.  Our data show, in this study, more significant effects when cells are treated with CBD alone. Therefore, we suggest that CBD alone could be used as both a novel treatment for NSCLC, but importantly, upon the development of resistance to cisplatin-based chemotherapy, our data suggest that CBD may play an important therapeutic role in overcoming this drug resistant phenotype by inhibiting the survival of resistant clones, increasing tumor cell apoptosis, downregulating cancer stemness and reducing tumor growth and metastasis. We have now addressed this in the revised version of the manuscript online 176 of the results sections.

Very minor irritations and “petty comments”

  1. Line 49 – my pdf from cancers produced a font change in the part of the text to large, bold.

Response: We have changed the font size throughout the manuscript to ensure that the font size is similar.

  1. Reference 15 - I fail to see the connection of reference 15 to the H&E staining. Also subtitle of the section should not be an abbreviation, I know it is common when referring to such a gold standard technique but should be explained a little more fully in publications

Response: We have corrected the reference in this section of the Materials & Methods (section 2.9) to correspond with reference (16) which includes the H&E staining protocol used in the current study. This has been updated in the reference/bibliography sentience We have also changed the subtitle to the full name of the technique used.

  1. For your information the special character “µ” is freely available in most fonts using alt 230. It has been used several times before in the text so I see no reason for using “uM” as a substitute for µM.

 Response: We have now replaced “uM” with “µM” throughout the manuscript.

  1. Line 155 – Tranilast – where did you get it and what concentration did you use, also the citations you use for TRPV2 in the introduction should at the very least include!

Response: We have added information of the company from which the Tranilast was purchased in addition to the concentration used in this study (section 2.10). In addition, we have included the following reference.

Perálvarez-Marín A, Doñate-Macian P, Gaudet R. What do we know about the transient receptor     potential vanilloid 2 (TRPV2) ion channel?. FEBS J. 2013;280(21):5471-5487. doi:10.1111/febs.12302

  1. Line 226 – grammar mistake should “maybe” not “may by”

Response: We have changed this in our revised manuscript in the line 232.

  1. Figure 6 is missing from my pdf printout, the caption is present though

Response: We updated the manuscript including some minor changes in the figure’s panels. We think therefore you may have received an older version for the manuscript. The revised updated version now includes figure 6.

  1. Line 385 Sentence doesn’t quite make sense, I think you left out an “in”

Response: We have now corrected this error and have revised the manuscript to include this in line 385.

Reviewer 4 Report

In this paper the authors tested a hypothesis that a non-psychotropic phytochemical, Cannabidiol (CBD) inhibits growth and metastasis of drug-resistant NSCLC cells in vitro and in vivo. Authors have previously demonstrated that CBD significantly increased uptake of cytotoxic chemotherapy Doxorubicin by the triple-negative breast cancer cells in part through agonism of an ion channel Transient receptor potential vanilloid type-2 (TRPV2). Here the authors utilize cisplatin-resistant NSCLC cells and show that CBD affected viabilities of parental (PT) and Cisplatin-resistant (CR) NSCLC cells. CBD also impacted abilities of CR cells to form colonies in suspension. CBD-dependent inhibition of CR cells involved elevated levels of ROS, and activation of intrinsic apoptosis signaling. CBD treatments also inhibited 3d growth of CR cells as spheres and down-regulated various invasion and metastasis promoting genes. Last but not least, CBD inhibited growth of CR-cell derived tumors in NSG mice in part through activating apoptosis and suppressing micro-metastases. Authors also indicate TRPV2 expression is observed in lung cancer patients and that higher TRPV2 expression correlates with better overall survival. Moreover, use of TRPV2 inhibitor indicated a requirement of this ion channel in transducing inhibitory effects of CBD in CR cells. Although the results indicate figure 6 legend, the figure 6 data are missing.  The studies overall are well conducted with appropriate data analyses with necessary statistics. Authors will need to revise this paper to include figure 6 data and perform further editorial corrections for typos and enhance comprehension. Authors are also requested to clarify whether CBD induced levels of apoptosis signaling in PT versus CR cells were similar.

Author Response

As requested by the Editor and based on the reviewers comments below, we have amended our manuscript in accordance with these suggestions in this most updated revised version. We trust that these minor corrections are now adequate to warrant publication in the journal, Cancers.

Comments and Suggestions for Authors

In this paper the authors tested a hypothesis that a non-psychotropic phytochemical, Cannabidiol (CBD) inhibits growth and metastasis of drug-resistant NSCLC cells in vitro and in vivo. Authors have previously demonstrated that CBD significantly increased uptake of cytotoxic chemotherapy Doxorubicin by the triple-negative breast cancer cells in part through agonism of an ion channel Transient receptor potential vanilloid type-2 (TRPV2). Here the authors utilize cisplatin-resistant NSCLC cells and show that CBD affected viabilities of parental (PT) and Cisplatin-resistant (CR) NSCLC cells. CBD also impacted abilities of CR cells to form colonies in suspension. CBD-dependent inhibition of CR cells involved elevated levels of ROS, and activation of intrinsic apoptosis signaling. CBD treatments also inhibited 3d growth of CR cells as spheres and down-regulated various invasion and metastasis promoting genes. Last but not least, CBD inhibited growth of CR-cell derived tumors in NSG mice in part through activating apoptosis and suppressing micro-metastases. Authors also indicate TRPV2 expression is observed in lung cancer patients and that higher TRPV2 expression correlates with better overall survival. Moreover, use of TRPV2 inhibitor indicated a requirement of this ion channel in transducing inhibitory effects of CBD in CR cells. Although the results indicate figure 6 legend, the figure 6 data are missing.  The studies overall are well conducted with appropriate data analyses with necessary statistics. Authors will need to revise this paper to include figure 6 data and perform further editorial corrections for typos and enhance comprehension.

Authors are also requested to clarify whether CBD induced levels of apoptosis signaling in PT versus CR cells were similar.

Response: We thank Reviewer 4 for these insightful comments. We apologize for this error in Figure 6 of the original manuscript. The current revised manuscript has now been updated to include Figure 6. We have previously shown that CBD has similar effects on cell viability in both PT and CR cells (figure 1). Since our manuscript was focused primarily on the effect of CBD on the more aggressive cisplatin resistant phenotype in NSCLC cells, we did not evaluate the apoptosis signaling in PT cells in the presence of CBD treatment as part of this study.